# *In situ* Conservation Assessment of Forage and Fodder CWR in Spain Using Phytosociological Associations

**María Luisa Rubio Teso \* and José M. Iriondo \***

Department of Biology and Geology, Physics and Inorganic Chemistry, Biodiversity and Conservation Area, University Rey Juan Carlos, 28933 Móstoles, Spain

\* Correspondence: marialuisa.rubio@urjc.es or mluisarubioteso@gmail.com (M.L.R.T.); jose.iriondo@urjc.es (J.M.I.)

**Abstract:** Crop wild relatives (CWR) can be used to mitigate the negative effects of climate change on crops, but their genetic diversity conservation has not been properly addressed. We propose a new target unit for conservation (Asso-EcoU) based on the occurrence of phytosociological associations in different environments. This approach involves using ecogeographical information and distribution data of associations to identify an optimized set of locations for *in situ* genetic diversity conservation. Thirty-nine CWR species, grouped in 15 associations, generated 165 Asso-EcoUs. Using the Sites of Community Importance (SCI) of Natura 2000 in Spain, we performed three analyses: (1) gap analysis, (2) coverage of the network, and (3) complementarity analysis. Analyses were performed with both target conservation units, associations, or Asso-EcoUs. The SCI network includes 100% of the associations and 69% of the Asso-EcoUs. The coverage assessment showed that 8.8% of the network is enough to encompass all the networks' target conservation units. Complementarity analysis showed that seven and 52 SCI areas are needed to contain at least one site of the 15 associations and the 114 Asso-EcoUs, respectively. These results highlight the value of Asso-EcoUs to potentially incorporate the genetic diversity component into conservation plans, while increasing the number of species covered.

**Keywords:** CWR; optimized conservation; phytosociological associations; Ecogeographical Land Characterization (ELC) maps; genetic diversity; Sites of Community Importance (SCI) sites; Natura 2000

## 1. Introduction

Crop Wild Relatives (CWR) are wild species that are closely related to crop species to which they contribute genetic material [1]. According to Lidder & Sonnino [2], CWR have more genetic diversity than crops, which are known to have a narrow genetic base as a result of domestication and genetic breeding, especially in recent decades [3]. This greater diversity and the environmental pressures to which they are subjected provide them with specific adaptive traits [4,5] that can be used to improve crop characteristics. In fact, they have already been successfully used in breeding programs of many crops [6–8]. Thus, CWR should be considered essential actors in sustaining food security and providing the adaptations needed to face the challenges brought about by climate change in the coming years.

CWR genetic diversity conservation has traditionally been neglected [8]. Genetic erosion as well as habitat fragmentation including changes in land use and competition with alien species have been identified as the main agents threatening their integrity [9]. Nevertheless, in recent years, important steps have been taken to protect CWR as a valuable component of plant diversity. CWR have been

included in National Plans for Plant Conservation [10] and national prioritized checklists of CWR and strategies for their conservation have been developed [11]. However, the listing of CWR species for their conservation may be insufficient if the main target is to preserve their genetic diversity. *Ex situ* conservation of CWR in gene banks is the most straightforward approach already pursued by several countries [11]. Nevertheless, it needs to be complemented with the *in situ* conservation of natural populations, where genetic diversity is dynamically evolving in response to an environmental change.

Although some studies have proposed certain areas for establishing CWR genetic reserves [12,13], there are still few comprehensive approaches to conserve *in situ* the genetic diversity of a targeted set of priority CWR species in a territory [see 14]. Adequately assessing genetic diversity by DNA characterization of all populations of a set of targeted CWR species is currently unfeasible due to limited economic resources. Consequently, alternative methods that capture the specific environmental attributes of the area under study can be used as a proxy of genetic diversity in order to identify possible adaptation patterns of plant populations [15,16]. One of them involves the use of ecogeographical land characterization maps (ELC maps) [15]. Heterogeneous environmental conditions are expected to generate diverging selective pressures and, therefore, generate genetic differentiation of adaptive value. Thus, the classification of the variety of habitats found in a territory with ELC maps can be used to identify natural populations with potentially different adaptations [17]. In any case, when specific traits are sought, the use of ecogeographical information as a proxy for adaptive genetic differentiation among populations must be subsequently validated by appropriate phenotype evaluation and/or molecular marker characterization of the target populations.

Recent research has explored the use of ELC maps to study the distribution of a group of given CWR species along the ecogeographical categories of a given territory [18–20]. However, these studies do not explicitly link each ecogeographical category (hereafter, ELC categories) to each population. A new conservation unit based on each of the different ELC categories in which a species is distributed could be considered a valid approach to identify the genetic differentiation of adaptive values among populations that can be found in a species. However, this approach leads to a considerable increase in the number of conservation targets (target species x ecogeographical categories in which they are found). Thus, in this sense, it would be an advantage in terms of time, cost, and resources to manage the conservation of genetic diversity of various CWR species simultaneously.

The study of plant communities and aggregation of species in repetitive patterns considering their relationships and environmental dependence [21] may provide an alternative approach to reduce the number of target conservation units. The classification of phytosociological associations, known as syntaxonomy, is based on inventories listing all plant species co-occurring in an area. Thus, the basic syntaxonomical units – associations – are identified by comparing the inventories and fidelity of the species to a given plant community. In this context, the use of the association as a management unit could be used to conserve several CWR species that co-occur in a given set of environmental conditions. In this way, although the incorporation of ecogeographical categories as a proxy for genetic diversity conservation would increase the number of target units for conservation, the use of associations rich in target species could be helpful to manage several species at the same time. Forage and fodder CWR is an important group of CWR formed by species that commonly occur together in natural grasslands. Thus, they are especially suited for testing an association mediated *in situ* conservation approach.

The objective of this paper is to assess the *in situ* conservation of forage and fodder CWR in Spain by taking into account the ecogeographic heterogeneity of their populations. The assessment is based on the protection provided by the Sites of Community Interest (SCI) of Spain belonging to the Natura 2000 network. The conservation strategy proposed is built on the use of selected phytosociological associations containing CWR species. It is further based on the use of ELC categories (ecogeographical units in which a territory is divided) as a proxy of genetic diversity. For this purpose, we generated a new target conservation unit composed of the combination of

phytosociological associations that contain targeted CWR and ELC categories of a territory where these associations occur, which we named Asso-EcoU. As part of the assessment, we posed the following questions: 1) Are phytosociological associations a valid approach for *in situ* conservation of fodder and forage CWR? 2) How well are the target Asso-EcoUs and associations of fodder and forage CWR protected by the SCI of the N2000 network? 3) Which would be the priority sites to establish genetic reserves to actively conserve *in situ* this group of CWR? To answer these questions, we carried out gap, coverage, and complementarity analyses and used the concept of effectiveness, as described by Caro [22]. One of our main concerns was to use a proxy to incorporate the genetic differentiation between the populations component when planning conservation actions for multiple species in protected areas. We found that the use of phytosociological associations combined with ecogeographical information of the territory could be an efficient way to manage and conserve *in situ* various target forage and fodder CWR species at the same time, including a representative sample of their genetic differentiation of adaptive value among populations. This makes an efficient use of existing conservation resources.

## 2. Materials and Methods

### 2.1. Selection of Species and Phytosociological Associations

Target CWR species were selected from the Prioritized Spanish Checklist of crop wild relatives [23], among the subgroup category Fodder & Forage. According to the genepool concept of Harlan and de Wet [24], which classifies wild plant species according to their crossability to crops. We selected Fodder & Forage CWR species belonging to genepool 1B (same species as the crop species).

The resulting 45 species belonging to GP 1B were then introduced into the SIVIM database (http://www.sivim.info/sivi/, last accessed 05/07/2019) to obtain a list of the phytosociological associations where at least two of the targeted CWR were found. The first three associations or sub-associations, to which each target CWR species was more prevalent (higher fidelity), were selected. Those associations that were repeated two or more times in the previous selection were chosen as target associations to conserve and their corresponding distribution data were downloaded from SIVIM (Iberian and Macaronesian Vegetation Information System) (http://www.sivim.info/sivi/, last accessed 25/08/2019). Occurrence data were transformed from Military Grid Reference System coordinates into geographic decimal coordinates using MSP Geotrans software v. 3.5, developed by the National Geospatial-Intelligence Agency of the United States (https://earth-info.nga.mil/GandG/update/index.php?action=home, last accessed 30/07/2019). To ensure that the target species were present in the selected association, we only used inventories of the association while explicitly citing the occurrence of the target species. In addition, we only used data records that provided an accuracy of at least 1x1 km.

### 2.2. Ecogeographical Analyses

#### 2.2.1. Creation of the Ecogeographical Land Characterization Map and Representativeness Analysis

An Ecogeographical Land Characterization map was created using the ELCmap tool of Capfitogen software v. 2.0 [25]. Its extent was restricted to Peninsular Spain and the Balearic Islands because none of the selected associations were found in the Canary Islands. The *kmeanbasic* function was used to define the optimal number of ELC categories for the ELC map. This function uses a clustering algorithm determining the cut-off points based on the decrease in the sum of the intra-group squares. When the decrease in the intra-group sum of squares in a range of n and n+1 groups is less than 50%, the algorithm reaches the optimal number of categories [26]. It has been successfully used for the creation of other ELC maps, as described in Reference [17]. The territory was structured using a grid of 1x1 km cells. The ecogeographical variables used to create the ELC map were: (i) climatic variables: annual mean temperature, annual precipitation, temperature seasonality, temperature annual range, (ii) geophysic variables: elevation, slope degrees, (iii) soil variables: topsoil clay fraction, topsoil salinity, topsoil sodicity, topsoil organic carbon, topsoil pH,

topsoil sand fraction, topsoil slit fraction, and (iv) other variables: latitude and longitude, which were included to create spatially aggregated categories. All these variables were chosen, in agreement with expert advice, to consider relevant ecogeographical factors in species distribution, which aim to create a generalist ELC map that would discriminate different adaptive environments for species with diverse ecological requirements.

The Capfitogen Representa tool of Capfitogen software v 2.0 was used to incorporate into the association occurrences, where the corresponding ELC category was derived from the ELC map of Spain with a 1x1 km resolution. The Asso-EcoU conservation unit represents those sites where a specific phytosociological association occurs under a particular ELC category. Thus, each combination of a selected phytosociological association with each of the ELC categories in which it is distributed becomes a unique Asso-EcoU.

## 2.2.2. Gap, Coverage, and Complementarity Analyses

Gap analyses [27] were carried out to estimate the protection provided by the SCI network of the Natura 2000 network. Thus, through gap analyses, we determined the effectiveness of the network, i.e., what percentage of the targeted associations and Asso-EcoUs were found at least once within the limits of the Sites of Community Importance of the Natura 2000 network in Spain. On the other hand, through coverage analysis, we determined the efficiency of the SCI network. The ratio of the number of protected areas containing the targeted conservation units versus the total number of protected areas of the network [25].

A complementarity analysis was carried out to identify the set of areas that encompass the maximum number of target conservation units in the minimum number of sites, following an iterative process [28]. First, the area containing the greatest number of different target conservation units is selected. Then, those target conservation units already present in the first area are excluded from the analysis and a second area containing the greatest number of possible different target conservation units is chosen. This process is repeated until all target conservation units under analysis are covered.

All the analyses were performed with the associations and Asso-EcoUs, using the SCI network of Spain (Natura 2000 network) as target protected areas. The latter were downloaded from: https://www.miteco.gob.es/es/biodiversidad/servicios/banco-datos-naturaleza/informacion-disponible/rednatura_2000_lic_descargas.aspx# (last accessed 22/08/2019). The Complementa tool developed under Capfitogen software v 2.0 was used to perform all these analyses.

## 3. Results

### 3.1. Selection of Species and Phytosociological Associations

The selection of three associations for each of the 45 target forage and fodder CWR species resulted in a total of 84 different associations or sub-associations. Two species (*Lupinus consentinii* Guss. and *Trifolium vesiculosum* Savi) were not found in any of the inventories of the phytosociological associations recorded in the SIVIM database. Table S1 shows the list containing the three associations per target CWR species, the number of inventories of each association, the number of inventories in which the species is cited, and the fidelity of the target CWR species of the association.

Of these 84 associations, 21 were initially selected for analyses since they were repeated at least two times (see Table S2). The data quality filtering procedure caused the exclusion of six phytosociological associations. This resulted in 15 associations containing 39 of the initial 45 species (Table 1).

**Table 1.** List of the 15 phytosociological associations chosen as a conservation target containing 39 priority forage and fodder crop wild relative (CWR) species of Spain. The number of inventories in which each CWR species is present in the association and the fidelity to the association are shown.

Inventories: Number of inventories in which the species is present. Fidelity (%): percentage of inventories of the association in which the species is present.

| | Target Associations | Priority CWR Species | Inventories | Fidelity (%) |
|---|---|---|---|---|
| 1 | *Agrimonio-Trifolietum medii subass. primuletosum columnae* | *Astragalus glycyphyllos* L. | 10 | 19.6 |
| | | *Trifolium medium* L. | 44 | 86.3 |
| 2 | *Euphrasio-Plantaginetum mediae* | *Poa alpina* L. | 32 | 11.9 |
| | | *Poa compressa* L. | 12 | 4.5 |
| | | *Agrostis capillaris* L. | 121 | 45.0 |
| 3 | *Festuco amplae-Agrostietum castellanae* | *Deschampsia cespitosa* (L.) P. Beauv. | 6 | 1.9 |
| | | *Trifolium dubium* Sibth. | 87 | 27.9 |
| 4 | *Gaudinio verticicolae-Hordeetum bulbosi* | *Hedysarum coronarium* L. | 21 | 95.5 |
| | | *Trifolium lappaceum* L. | 1 | 4.5 |
| | | *Trifolium squamosum* L. | 9 | 40.9 |
| | | *Trifolium squarrosum* L. | 2 | 9.1 |
| 5 | *Helianthemetum guttati* | *Medicago truncatula* Gaertn. | 9 | 8.7 |
| | | *Trifolium nigrescens* Viv. | 9 | 8.7 |
| 6 | *Holoschoenetum vulgaris* | *Agrostis stolonifera* L. | 214 | 41.2 |
| 7 | *Linario eleganti-Anthoxanthetum aristati* | *Lupinus luteus* L. | 7 | 14.3 |
| | | *Medicago arabica* (L.) Huds. | 1 | 2.0 |
| | | *Ornithopus sativus* Brot. | 5 | 10.2 |
| | | *Lupinus angustifolius* L. | 8 | 16.3 |
| 8 | *Lolio-Plantaginetum majoris* | *Lolium perenne* L. | 308 | 88.3 |
| 9 | *Medicagini rigidulae-Aegilopetum geniculatae* | *Medicago rigidula* (L.) All. | 150 | 38.4 |
| 10 | *Rhinantho mediterranei-Trisetetum flavescentis* | *Dactylis glomerata* L. | 117 | 95.9 |
| | | *Medicago lupulina* L. | 70 | 57.4 |
| | | *Trifolium incarnatum* L. | 2 | 1.6 |
| | | *Trifolium pratense* L. | 113 | 92.6 |
| | | *Festuca arundinacea* Schreb. | 16 | 13.1 |
| | | *Festuca pratensis* Huds. | 39 | 32.0 |
| | | *Medicago sativa* L. | 23 | 18.9 |
| | | *Poa pratensis* L. | 70 | 57.4 |
| | | *Trifolium repens* L. | 78 | 63.9 |
| 11 | *Trifolio cherleri-Taeniatheretum caput-medusae* | *Trifolium arvense* L. | 87 | 52.9 |
| | | *Trifolium angustifolium* L. | 70 | 45.8 |
| | | *Trifolium campestre* Schreb. | 89 | 58.2 |
| | | *Trifolium striatum* L. | 56 | 36.6 |
| 12 | *Trifolio fragiferi-Cynodontetum dactyli* | *Medicago scutellata* (L.) Mill. | 1 | 0.3 |
| 13 | *Trifolio subterranei-Poetum bulbosae* | *Trifolium resupinatum* L. | 4 | 7.4 |
| | | *Poa bulbosa* L. | 54 | 100.0 |
| | | *Trifolium subterraneum* L. | 31 | 57.4 |
| 14 | *Trifolio cherleri-Plantaginetum bellardii* | *Ornithopus compressus* L. | 81 | 53.6 |

| 15 | *Lino biennis-Cynosuretum cristati* | *Lolium multiflorum* Lam. | 22 | 5.9 |
| --- | --- | --- | --- | --- |

### 3.2. Ecogeographical Analyses

3.2.1. Creation of Ecogeographical Land Characterization Map and Representativeness Analysis

The clustering algorithm using the selected environmental variables resulted in an ELC map with 49 different ELC categories. The total number of inventories of the selected associations was 1283, distributed across 39 ELC categories. Inventories found to be in ELC categories '0' or 'NA' (no available information) were excluded from the analysis, which resulted in 1227 total inventories. The distribution of the inventories of the associations along the ELC categories depended on the frequency of the ELC categories in the territory and the sampling effort (Table S3). The associations that were found in a greater number of ELC categories were *Holoschoenetum vulgaris* and *Medicagini rigidulae-Aegilopetum geniculatae*, which also had the highest number of inventories. On the other hand, two other associations (*Lino biennis-Cynosuretum cristati* and *Festuco amplae-Agrostietum castellanae*) with a similar sampling effort had a low number of ELC categories, which relate to more specific-habitat requirements (Figure 1).

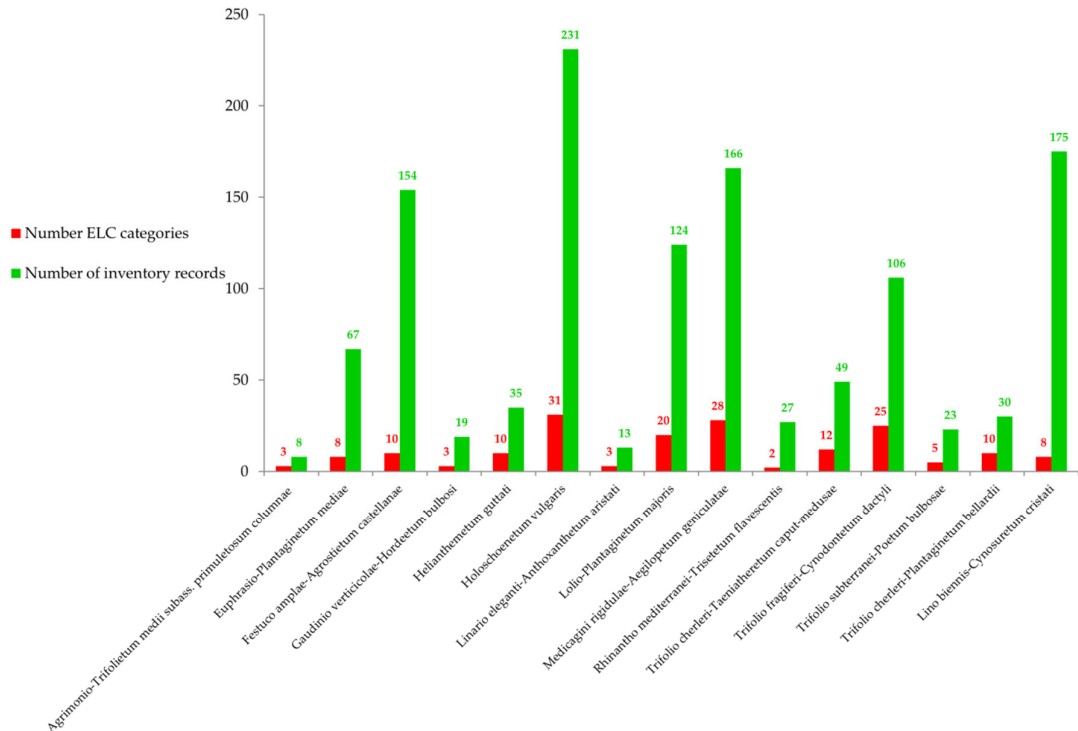

**Figure 1.** Number of inventory records of the target phytosociological associations containing prioritized species of forage and fodder crop wild relatives of Spain and the number of Ecogeographical Land Characterization categories represented in the inventories of each association.

Lastly, the combination of the 39 ELC categories with the occurrence data of the 15 phytosociological associations resulted in 165 unique Asso-EcoUs. Figure 2 shows a representation of the ELC map with the 165 Asso-EcoUs distribution.

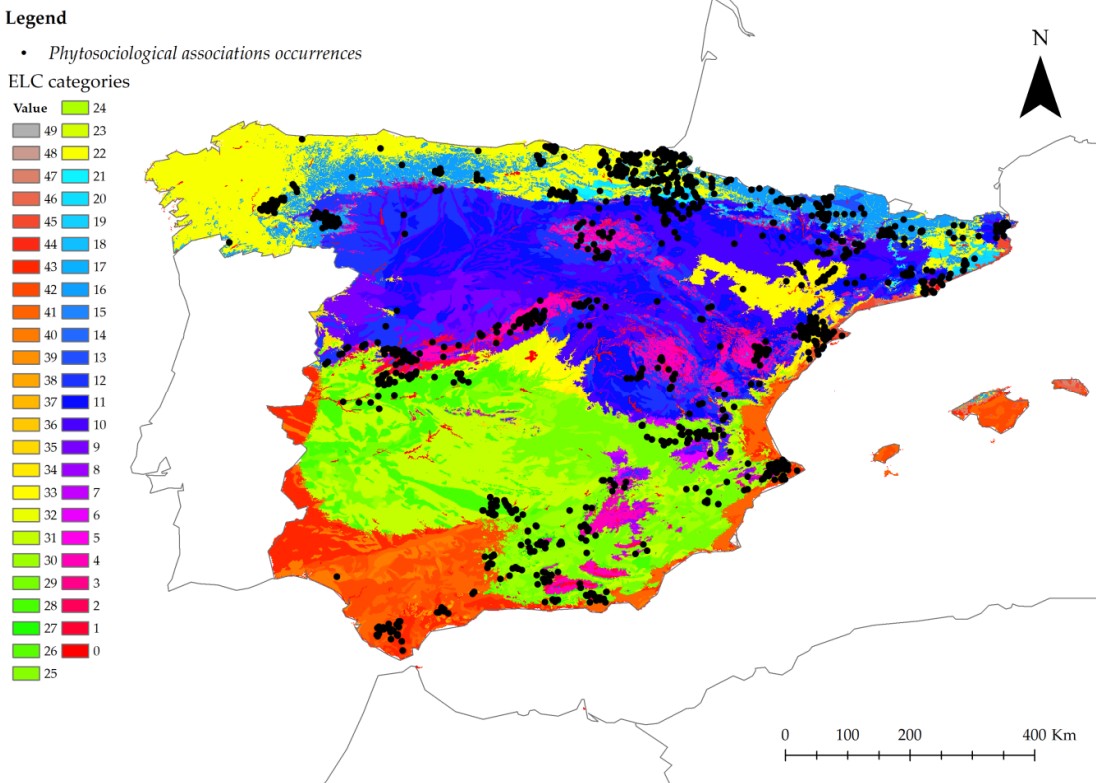

**Figure 2.** Ecogeographical Land Characterization (ELC) map of Spain composed of 49 ecogeographical categories. Black dots depict the occurrences of 1227 inventories representing 165 Asso-EcoUs (combination of associations with ELC categories) (15 phytosociological associations that contain 39 priority forage and fodder crop wild relative species).

### 3.2.2. Gap Analysis, Coverage, and Complementarity Analyses

The gap analysis showed that 447 of the 1227 inventories were found within the SCI areas of the Natura 2000 network, representing at least once, all 15 associations (100%) (Table S4) and 114 of the 165 targeted Asso-EcoUs (69%) (Table S5).

The Natura 2000 network of Spain contains 1449 SCI areas and the coverage analysis showed that 127 SCI areas will suffice to passively protect the 15 associations and the 114 Asso-EcoUs present in the network. These 127 areas represent 8.8% of the total areas composing the SCI network of Natura 2000 in Spain, which points to a low efficiency of the network in covering our target conservation units. Table S6 shows the 127 SCI areas covering the targeted conservation units. It is worthy to note that 37 areas have a greater number of Asso-EcoUs inventories than phytosociological associations inventories, which indicates that these associations are found under diverse environmental conditions in the same SCI area.

The complementarity analysis showed that seven SCI areas (0.48% of the network) would be needed to conserve all 15 associations. In addition, these seven areas would contain 22 different Asso-EcoUs (17.5% of Asso-EcoUs inside SCI areas) (Table 2). Same analysis performed with the 114 Asso-EcoUs showed that 52 SCI areas (3.59% of the network) is the minimum number of areas to represent Asso-EcoU at least once each (Table S7). Figure 3 shows the maps representing the SCI areas obtained from complementarity analysis targeting the associations or the Asso-EcoUs. Fourteen SCI areas add at least three new different Asso-EcoUs to the sites selected through complementarity analysis, which represents around 41% of total Asso-EcoUs (67 different Asso-EcoUs) and 14 of the 15 associations. To complete the coverage of the associations and include the missing association, (*Linario eleganti-Anthoxanthetum aristati*). The SCI area named Baixa Limia must be added since it is the only one including that association (see Table S4). These 15 areas

constitute our proposal for establishing genetic reserves of phytosociological associations containing prioritized forage and fodder CWR of Spain (Figure 4). This proposal includes all the targeted associations and 41% of the targeted Asso-EcoUs (68 different Asso-EcoUs). Both results using the associations or the Asso-EcoUs point at a high effectiveness of the network in covering our target conservation units.

**Table 2.** Areas belonging to the Sites of Community Importance of Spain that will cover the 15 selected associations under study obtained through complementarity analysis.

| Name of SCI Area | Site Code | Autonomous Community | Number of New Associations Added | Number of Asso-EcoUs Present |
|---|---|---|---|---|
| Cuenca del río Lozoya y Sierra Norte | ES3110002 | Comunidad de Madrid | 5 | 9 |
| Aiguamolls de l'Alt Empordà | ES0000019 | Cataluña | 3 | 3 |
| Ordesa y Monte Perdido | ES0000016 | Aragón | 3 | 3 |
| Los Alcornocales | ES0000049 | Andalucía | 1 | 4 |
| Montaña Oriental | ES1300002 | Cantabria | 1 | 1 |
| Baixa Limia | ES1130001 | Galicia | 1 | 1 |
| Montes Aquilanos y Sierra de Teleno | ES4130117 | Castilla y León | 1 | 1 |

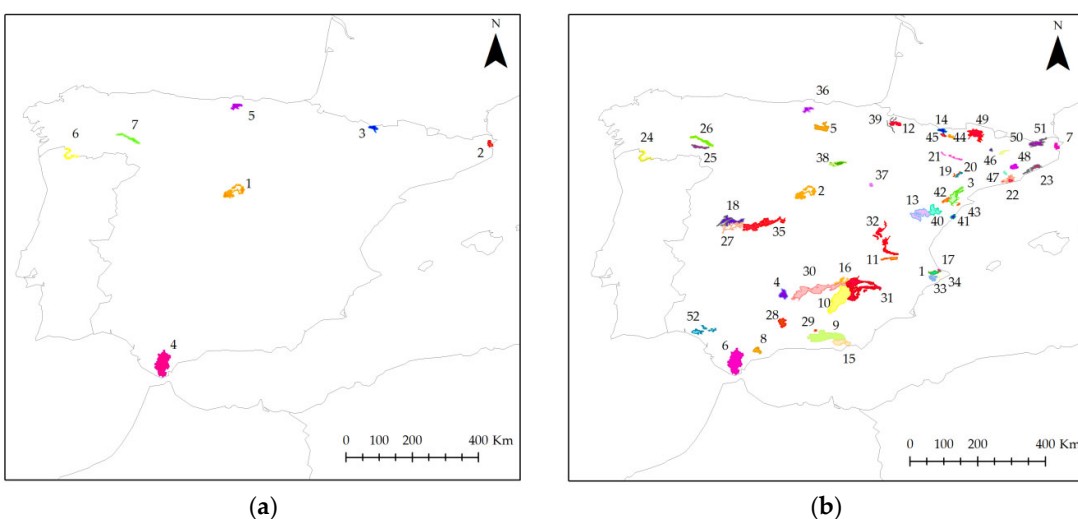

**Figure 3.** Graphic comparison between the seven Sites of Community Importance (SCI) areas needed to conserve the 15 phytosociological associations (a) vs. the 52 SCI areas needed to conserve the 114 Asso-EcoUs (combination of associations with ecogeographical categories) under study (b). The selection of areas was made through complementarity analysis and numbers by the SCI areas indicate the ranking in the selection process.

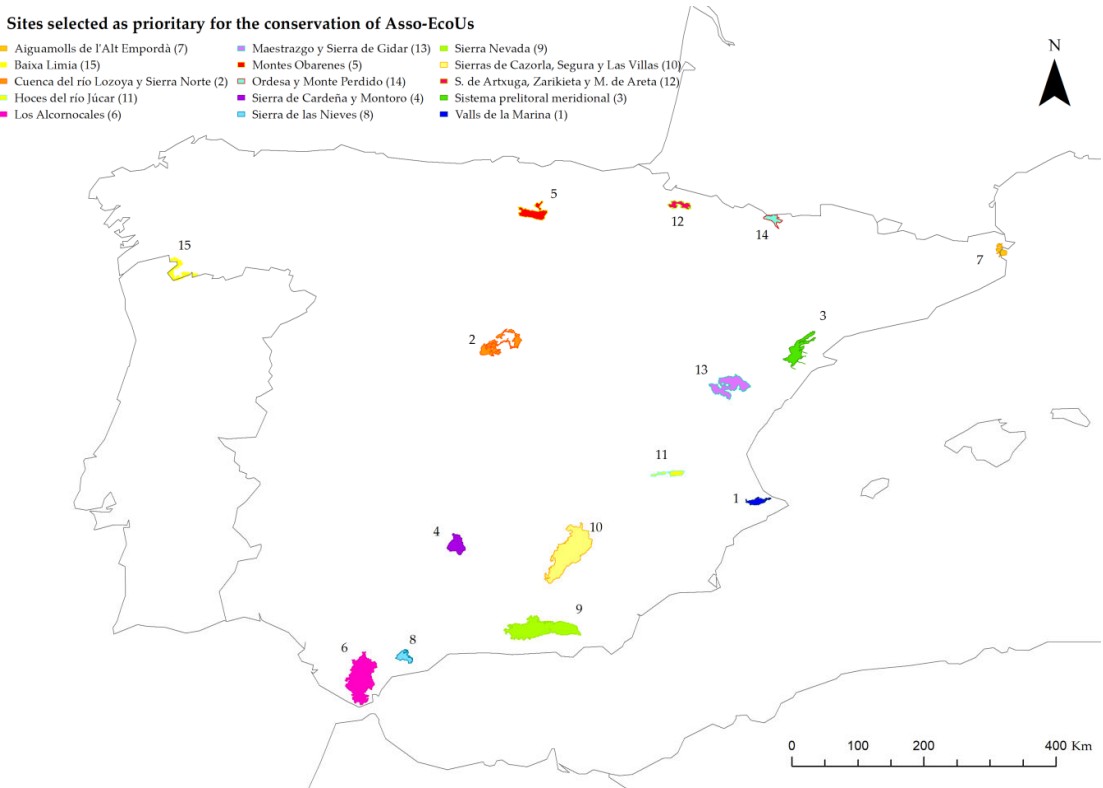

**Figure 4.** Proposal of 15 Sites of Community Importance (SCI) areas of the Natura 2000 network to establish genetic reserves for the conservation of prioritized forage and fodder crop wild relatives of Spain grouped in Asso-EcoUs. Selection of SCI areas was made through complementarity analysis, and the numbers by the SCI areas indicate the ranking in the selection process.

## 4. Discussion

### 4.1. Relevance of Genetic Diversity in the Conservation of Crop Wild Relatives

The preservation of target CWR species, which protects both their taxonomic diversity and genetic diversity, should be one of the objectives when planning CWR conservation strategies, especially at the national level [29]. Traditionally, the steps taken in the implementation of national CWR strategies include the creation of checklists listing all CWR species of a country followed by their prioritization, according to different criteria depending on the country's idiosyncrasy [14]. However, in several cases, the national CWR conservation strategies do not go much further and do not take practical steps for the conservation of a representative sample of the genetic diversity of adaptive value present in each CWR. As a result, the genetic diversity of the target species may not be conserved or it is, at most, conserved in an inefficient manner.

One of the most important characteristics of CWR is that their populations can provide useful genes to their domesticated relatives [4,30]. Thus, CWR are an important source to take into account in crop breeding [7,31] given the expected negative effects of climate change on crop yields [32] and the inability of crops to overcome extreme events due to their narrow genetic diversity [3]. Thus, conservation goals should not be focused on the conservation of the target CWR species, but on the conservation of several populations per species containing a representative sample of their genetic diversity of adaptive value.

To that aim, maps that capture the specific environmental attributes of an area (ELC maps are built considering climatic, edaphic, and geophysical variables) [15] can be used as a proxy to survey the genetic differentiation of adaptive value among CWR populations [16]. Consequently, they have already been used for this purpose in several cases, such as in the improvement of seed collections of

germplasm banks [33], the search for tolerance to drought and salinity in *Aegilops* collections of Spain, by taking into consideration different adaptive contexts [17] or the assessment of CWR diversity in Spain [19] and Norway [18]. When this approach is used, results must be validated by proper phenotyping and/or molecular characterization procedures.

A further step forward in this line involves considering each potentially different adaptive environment of a CWR as a conservation target. However, one of the drawbacks of this approach is that the number of conservation targets may become too large to handle. Hence, new approaches to achieve the *in situ* conservation of CWR in an efficient manner are needed.

### 4.2. Phytosociological Associations as a Means for Multiple Species Conservation Management

Phytosociology, as a discipline with a long tradition in Europe, facilitates the management of various species at a time – a desirable characteristic for conservation actors – using the association as a reference entity. This can be especially advantageous if the objective is to conserve species and their genetic diversity while considering the ecogeographical information associated with them [34]. The link of ecological information with phytosociological associations has been previously approached from a landscape perspective by proposing a landscape unit named *geosigmetum* [35], which consists of an integrated set of vegetation series that is repeated in an area of land with the same edaphic, climatic, and biogeographic characteristics. On the contrary, in our case, we are interested in the different environmental conditions that can be found within the distribution range of an association and which can generate different evolutionary adaptations in their populations. Using the Asso-EcoU as target conservation units in CWRs can be helpful in establishing genetic reserves if there are several CWR species that form part of the same association. Fodder and forage CWR can be a good group to test the viability of this novel approach, since many of these species are known to grow together. Results in the selection of associations support this hypothesis since the associations contain from seven to 20 prioritized forage and fodder crop wild relatives, except for one that has three target species (see Table S2).

The viability of using the Asso-EcoU approach also depends on the quality and quantity of distribution data of the phytosociological associations. In this sense, we verified this premise by finding that most selected associations (15 of 21) had accurate georeferencing data at the 1x1 km grid cell scale. Considering that, even if the species is very frequently linked with the association, its presence in a citation of an association is not assured. It becomes necessary to use only inventory records of the association where the species has been cited. Alternatively, it is possible to cross the association distribution data with the target species distribution data obtained from an independent source (e.g., Global Biodiversity Information Facility (GBIF)) to verify the presence of the species at one location.

### 4.3. Ecogeographical Analyses

#### 4.3.1. Creation of ELC Map and Representativeness Analysis

The generalist ELC map created for forage and fodder crop wild relatives of Spain and used in this study resulted in 49 ELC categories, which were assigned to the territory with a 1x1-km grid resolution. The distribution of inventories throughout the ELC categories agrees in general terms with their expected distribution, according to the frequency of ELC categories in the ELC map of Peninsular Spain and Balearic Islands (see Table S3). ELC categories with low frequency occurrence in the studied territory have a lower number of inventory records than ELC categories with higher frequencies. Thus, nine of the 10 ELC categories that do not have any inventory records correspond to ELC categories with low representation in the territory studied (<0.03%). The exception is category 49, which has a mid-high distribution. The database of inventories of phytosociological associations of SIVIM provides a balanced sample of studies distributed across the different ELC categories. This screening also indicates that the set of phytosociological associations selected for this study cover almost all of the most frequent types of environmental conditions (ELC categories) found in Peninsular Spain and Balearic Islands.

4.3.2. Gap Analysis, Coverage, and Complementary Analyses

Gap analysis help answer the question of how our target conservation units – phytosociological associations or Asso-EcoUs – are passively protected by the Sites of Community Importance of the Natura 2000 network of Spain. The N2000 network was developed under the Habitats Directive seeking the protection of threatened habitats and species [36]. Results in this work point to a very high effectiveness (in terms of representation of all targeted CWR diversity) of the network, since all phytosociological associations and almost 70% of the Asso-EcoUs have at least one inventory within the network. This is congruent with previous studies performed on the Natura 2000 network in Spain in which they found a high effectiveness in covering threatened species, since only 5% of threatened species under study had less than 10% of their distributions not covered by the Natura 2000 network [37]. Other works, however, report low effectiveness of the Natura 2000 network concerning other parts of Europe or other groups of species, such as, plant regional biodiversity in Crete [38], terrestrial vertebrates and fresh water fishes in Italy [39], Mediterranean lichen species in Spain [40], or insects in Italy [41]. As pointed out by many of these authors, the adequacy of the Natura 2000 network to fulfill biodiversity conservation goals is heavily dependent on the taxonomic group under study and the heterogeneous contributions to the network made by the different countries, dependent on economic and political issues.

The use of existing protected areas should be the first approach for planning CWR *in situ* conservation actions since it minimizes costs and takes advantage of existing conservation laws and management figures. Nevertheless, there are some Asso-EcoUs not protected by the network that should also be included in a future network of genetic reserves. The absence of these Asso-EcoUs in the SCI network could be explained by two different hypotheses. The first one is that the ELC categories of those Asso-EcoUs are simply not found inside the limits of the SCI network. From our point of view, this is unlikely to happen, as the Natura 2000 network covers 118 different habitats in Spain, representing about 37% of the territory of the country (https://www.miteco.gob.es/es/biodiversidad/temas/espacios-protegidos/red-natura-2000/rn_espan a.aspx, last accessed 2/09/2019). In addition, the diversity of environments found within the limits of the SCI network is highlighted in previous works [18,19]. The second hypothesis is that the missing Asso-EcoUs are actually present in the SCI network but have not been captured by the current association inventories. We should exclude the distribution patterns of the associations as an explanation, as all associations are represented in the SCI network. In this sense, we recommend to prospect the missing associations in the SCI areas that contain the ELC categories corresponding to the targeted Asso-EcoUs.

If the missing targeted Asso-EcoUs cannot be found within the SCI network, the inclusion of new areas to formal networks of protected areas can be considered, which is a matter that has already been addressed [42]. In the case of *in situ* conservation of CWR, the existence of conservation targets outside protected areas is a common event and their management is an important matter that must be dealt with [43]. This intermediate solution would, thus, combine the network and other areas of interest, which could be designated micro-reserves or genetic reserves, as pointed out by Maxted et al. [16].

The coverage analysis denoted a low efficiency of the SCI network in terms of maximizing the use of resources of the network. With a low number of SCI areas (127 out of 1449), all targeted phytosociological associations or Asso-EcoUs inventories are passively protected. Again, this is not at all surprising since the Natura 2000 network was not designed to conserve forage and fodder CWR.

Complementarity analysis is a methodology largely discussed and evaluated through a comparison to other approaches [42,44–46]. It is widely used in plant conservation, including the conservation of crop wild relatives [47–50]. This methodology has been reported to be efficient for conservation purposes [51,52], even though, on the other hand, their weaknesses in the selection of areas have also been addressed [45,53]. As expected, the complementarity analysis performed with associations without considering ecogeographical information returned a significantly lower number of areas (SCI areas) needed for the preservation of all associations compared to the areas

needed to preserve all Asso-EcoUs (seven *vs.* 52 SCI areas of 1449 SCI areas). These results highlight that it is possible to generate an efficient subnetwork of genetic reserves covering all targeted conservation units in a low number of areas of the network. When we compare the target conservation units selected using associations and Asso-EcoUs in the SCI identified through complementarity analysis (see Table 2 and Table S7), it is possible to assess the relevance of the Asso-EcoU approach. The first selected SCI area using the associations' approach (Cuenca del río Lozoya y Sierra Norte in Comunidad de Madrid) includes five different target associations but nine target Asso-EcoUs, which validates the usefulness of the Asso-EcoUs when the goal is to maximize the conservation of populations subjected to different environmental pressures. Thus, the increase in the number of conservation targets is not a hindrance but an advantage, since it provides information on the potential adaptation patterns contained in the Asso-EcoUs occurrences that was not available with the associations' approach. The same pattern is followed by subsequent comparisons of complementarity SCI areas, which concurs with the reported usefulness of including environmental information to help increasing species-richness in complementarity areas [54]. It is in this context that Araujo & Williams [55] suggested incorporating ecological modelling information when applying complementarity algorithms, which, although increasing the number of areas to cover all species, provides higher probability of species persistence. In a nutshell, the combination of Asso-EcoUs as target conservation units and complementarity analyses provides a valid strategy for selecting areas to establish genetic reserves of crop wild relatives. The proposal of a subnetwork of 15 SCI areas to cover all targeted associations and 41% of the targeted Asso-EcoUs that are located inside the SCI network, should be seriously considered by biodiversity and plant genetic resources managers as potential sites to establish a long-term network of genetic reserves of forage and fodder crop wild relatives for Spain, where a representative sample of the genetic diversity of targeted CWR is likely to be captured.

Before any genetic reserve is established, the risk of outcrossing events with related crop species in surrounding areas should be evaluated and properly addressed. Furthermore, an experimental assessment of the adaptive traits of the selected populations should be performed if the aim is to consider these genetic reserves as an access point for using wild plant genetic resources for plant breeders, farmers, or other end users. Lastly, these genetic reserves should meet a minimum set of quality standards [56] to mitigate the effects of climate change and ensure the long-term conservation of these resources.

## 5. Conclusions

Phytosociological associations are a highly useful concept to facilitate the *in situ* conservation assessment of forage and fodder CWR species and identify an efficient network of sites for establishing genetic reserves. The incorporation of ecogeographical information to the selected associations as a proxy to represent potential genetic diversity of adaptive value among CWR populations by creating the Asso-EcoU target conservation unit, provides an inexpensive and simple approach for the *in situ* conservation of genetic diversity of several CWR species together. This is shown by the effectiveness of the SCI network in Spain in conserving the Asso-EcoUs and the possibility to establish an efficient network in a reduced number of sites obtained through complementarity analysis. The identification of suitable areas for implementing genetic reserves is more effective when using this new approach. In addition, the selection of areas through this method facilitates higher probability of species' persistence, since it is not a species, but the whole plant community that is managed for conservation. Notwithstanding these positive results, more research is still needed to assess the validity of this approach in other groups of CWRs and further efforts should be made to improve the coverage of association inventories throughout the territory.

**Supplementary Materials:** The following are available online at: https://github.com/MLRubioTeso/In-situ-conservation-assessment-of-forage-and-fodder-CWR-in-Spain-using-phytosociological-associatio/find/master?q=
Table S1: Table_S1_RubioTeso&Iriondo_2019_List_phytosociol_asso_Sustainability
Table S2: Table_S2_RubioTeso&Iriondo_2019_Phytosociol_asso_sel_Sustainability

**Author Contributions:** Conceptualization, M.L.R.T. and J.M.I. Methodology, M.L.R.T. and J.M.I. Software, M.L.R.T. Validation, M.L.R.T. Formal analysis, M.L.R.T. Investigation, M.L.R.T. Resources, J.M.I. Data curation, M.L.R.T. Writing—original draft preparation, M.L.R.T. Writing—review and editing, M.L.R.T. and J.M.I. Visualization, M.L.R.T. Supervision, J.M.I. Project administration, J.M.I. Funding acquisition, J.M.I.

**Funding:** The Horizon 2020 Framework Program of the European Union under grant agreement number: 774271 (Farmer's Pride project) funded this research and APC.

**Acknowledgments:** We thank Kurodo Kinoshita Kinoshita for his contribution in the initial stages of this study. We also thank Lori de Hond for her linguistic assessment. First author wants to specially thank Dr. Juan Antonio Sánchez Rodriguez, from University of Salamanca, who arose the curiosity in phytosociological associations in her.

**Conflicts of Interest:** The authors declare no conflict of interest. The funders had no role in the design of the study, in the collection, analyses, or interpretation of data, in the writing manuscript, or in the decision to publish the results.

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
