# Peer review of "In situ Conservation Assessment of Forage and Fodder CWR in Spain Using Phytosociological Associations"

_sustainability, doi:10.3390/su11215882_

Round 1

Reviewer 1 Report

This paper by Teso & Iriondo proposes a multi-step phytosociological method for the detection of areas with high conservation potential. Summarizing in very broad strokes, the method involves: i - choosing an appropriate set of eco-geographical variables; ii – create a geographic grid based on areas similar for the above set of variables; iii – a validation based on the comparison between the obtained grid and the actual position of target protected areas. Focus is on the identification of suitable conservation areas for maintaining wild relatives of crops (CWRs).

In the face of today climatic changes, the goal is a commendable one, also because, as the Authors point out, scarce attention has been given to CWR until now. Also, the method appears of broad application, even though the Authors limit themselves to the study of phytologic association for forage and fodder species.

I have only a concern, about the cavalier use that the Authors make of the term “genetic diversity” such as in the Abstract (“However, their genetic diversity conservation has not yet been properly addressed” – shaky English here, too) in the Introduction, culminating in the statement of line 80: “The objective of this paper is to assess the in situ conservation of genetic diversity of forage and fodder CWR in Spain”, in the Discussion (line 271: “… the use of ELC maps as estimators of genetic diversity.”) and in other passages. Alas, this paper DOES NOT assess genetic diversity and ELC maps DO NOT estimate genetic variability. Genetic variation can only be assessed by the use of genetic markers (phenotypic, molecular, whatever) and its distribution described by means of the models from Population Genetics theories (i.e. differentiation, structure etc.). None of this is presented in the work described in this paper and, in fact, any reference to genetic variation disappears in the “Conclusions” section. The examples I made must be considered conceptual mistakes. I agree that this kind of approach could very well complement genetic analysis, actually, I am looking forward to it, but the above should be clearly stated.

Therefore, I am asking for a minor revision aimed at removing all incorrect references to genetic variation from the text.

Author Response

Dear Editors,

Please find attached the revised version of our manuscript “In situ conservation assessment of forage and fodder CWR in Spain using phytosociological associations” for your consideration for publication as a research paper in Sustainability as part of the special issue entitled “Genetic Resources for Sustainable Agriculture”.

We have modified the original manuscript, taking into account all the reviewers’ comments. The main changes have been made in the introduction and discussion sections. Please find below our replies to the reviewers’ comments with a detailed explanation of the changes made. 

We thank the reviewers for their constructive feedback. We feel that the manuscript has substantially improved and hope that this version is suitable for publication. Both authors agree with the proposed changes.

Thank you for your time and consideration. We look forward to hearing from you.

Response to Reviewer 1 Comments

Point 1: I have only a concern, about the cavalier use that the Authors make of the term “genetic diversity” such as in the Abstract (“However, their genetic diversity conservation has not yet been properly addressed” – shaky English here, too) in the Introduction, culminating in the statement of line 80: “The objective of this paper is to assess the in situ conservation of genetic diversity of forage and fodder CWR in Spain”, in the Discussion (line 271: “… the use of ELC maps as estimators of genetic diversity.”) and in other passages. Alas, this paper DOES NOT assess genetic diversity and ELC maps DO NOT estimate genetic variability. Genetic variation can only be assessed by the use of genetic markers (phenotypic, molecular, whatever) and its distribution described by means of the models from Population Genetics theories (i.e. differentiation, structure etc.). None of this is presented in the work described in this paper and, in fact, any reference to genetic variation disappears in the “Conclusions” section. The examples I made must be considered conceptual mistakes. I agree that this kind of approach could very well complement genetic analysis, actually, I am looking forward to it, but the above should be clearly stated.

Response 1: 

We understand the reviewer’s concern on the application of the term ‘genetic diversity’ in the use of ELC maps. Indeed, the use of ELC maps is not a direct means to assess genetic diversity comparable to the use of genetic markers or the evaluation of phenotypes in common garden experiments. However, ecogeographic information can be used as a proxy of genetic differentiation generated by divergent natural selection. The basis of this approach is backed by the theory of population genetics: heterogeneous environmental conditions exert different selection pressures upon populations leading to genetic differentiation. In this context there are many studies that validate the use of this approach (e.g., Hedrick et al., 1976; Parra-Quijano et al., 2012, Preite et al., 2015, Egan et al., 2018, and references therein, among many others).

To attend the concerns of the reviewer and clarify this point, we have edited the abstract to improve the English expression. We have also edited the text in lines 58-69 (revised version) to fine tune the wording regarding genetic differentiation and to clearly state that this proxy must be subsequently validated using phenotypic evaluation and/or molecular characterization.

We have rewritten the objective of this paper omitting the word ‘genetic diversity’. Furthermore, we have reworded several sentences in the introduction and in the discussion (lines 87-88, 100-101, 105, 275-276, 280-281, 284, 384-385, in the revised version) eliminating the word ‘genetic diversity’ or substituting the term and evidencing the proxy character of this approach. In the discussion, we have reiterated the need to validate the results of this methodology with proper genetic analyses and phenotyping (line 280-281).

References:

Egan, P.A., Muola, A., Stenberg, J.A., 2018. Capturing genetic variation in crop wild relatives: An evolutionary approach. Evol. Appl. 1–12. https://doi.org/10.1111/eva.12626

Hedrick, P.W., Ginevan, M.E., Ewing, E.P., 1976. Genetic Polymorphism in Heterogeneous Environments. Annu. Rev. Ecol. Syst. 7, 1–32.

Parra-Quijano, M.; Iriondo, J.M.; Torres, E. Ecogeographical land characterization maps as a tool for assessing plant adaptation and their implications in agrobiodiversity studies. Genet. Resour. Crop Evol. 2012, 59, 205–217.

Preite, V., Stöcklin, J., Armbruster, G.F.J., Scheepens, J.F., 2015. Adaptation of flowering phenology and fitness-related traits across environmental gradients in the widespread Campanula rotundifolia. Evol. Ecol. 29, 249–267.

Reviewer 2 Report

All means that could simplify and optimize the uses of genetic resources are welcome for breeders.

Here for forage and fodder crops the Authors have based the strategy on the in situ way that is less costly that keeping plants in collections. Their goals are valuable.

However, the strategy developed by the authors is strictly theoretical and far from releasing genetic resources from wild accessions. Indeed, even wild located, these accessions were not evaluated and eventual gene flow with crops was not considered.

All the species are either outcrossed with wind or insect as pollinators that infers 5 km distance isolation from crops. Gene flow with crops are likely and authors should take into account in the database the isolation of the Natura 2000 sites. I am not Spanish and I cannot judge of this point for so many species.

Adaptation patterns are not so easy to find and I would be convinced that all species in the ELC show adaptation patterns for traits related to climate change. Salinity and drought traits do not share common patterns because drought stresses are very difficult to experiment in wild populations

Moreover, these genetic resources remain not evaluated and I cannot find in the text data on on their interest for breeding to climate change, except that there are wild and in Natura 2000 sites. These sites are frail and their evolution with climate change remains uncertain. In their sites the species cannot be evaluated that delays their use for eventual breeding. Climate changes are so rapid that I have doubts about the adaptation of wild species to survive in Natura 2000 sites.

In fact the authors should develop the benefit of a genetic resources in wild species located in an association versus genetic resources in collection. The interest of the associations is not so clear for me.

Either in the introduction or in discussion these points should be added

To summarize, in my opinion the strategy to make a database on wild forms on kinships species related to many crop species is publishable. But many cautions should be given and the concept of Asso-EcoU associations maintained in different environments, it should explain the relationships with adaptation patterns.

Please develop all sigles112 MGRS Military Grid Reference System

199 phytosociological associations resulted in 165 unique Asso-EcoUs Are there in fig. 2?

205 change ecogeographical categories.

In conclusion, the goals are fine, in my opinion the breeding objectives are too far, I would suggest to Authors to limit their ambition to the concept of Asso-EcoU not in relationships with climate changes because there are too much unknown relationships and that impair the studies.

In contrast, the evolution of associations in some sites basing on climate changes, obviously are long term studies... and eventually to introduce in the future studies neutral and linked molecular markers to salt and drought tolerance and other environmental traits to follow the evolution of markers that will indicate the fate of the traits.

Author Response

Dear Editors,

Please find attached the revised version of our manuscript “In situ conservation assessment of forage and fodder CWR in Spain using phytosociological associations” for your consideration for publication as a research paper in Sustainability as part of the special issue entitled “Genetic Resources for Sustainable Agriculture”.

We have modified the original manuscript, taking into account all the reviewers’ comments. The main changes have been made in the introduction and discussion sections. Please find below our replies to the reviewers’ comments with a detailed explanation of the changes made. 

We thank the reviewers for their constructive feedback. We feel that the manuscript has substantially improved and hope that this version is suitable for publication. Both authors agree with the proposed changes.

Thank you for your time and consideration. We look forward to hearing from you.

Response to Reviewer 2 Comments

Point 1: However, the strategy developed by the authors is strictly theoretical and far from releasing genetic resources from wild accessions. Indeed, even wild located, these accessions were not evaluated and eventual gene flow with crops was not considered

Response 1:

Certainly, the strategy developed is strictly theoretical and potential adaptative traits in selected sites should be confirmed by proper genetic analyses and phenotyping. In this sense, we have included in lines 61-63 the following sentence: “In any case, when specific traits are sought, the use of ecogeographic information as a proxy for adaptive genetic differentiation among populations must subsequently be validated by appropriate phenotype evaluation and/or molecular marker characterization of the target populations”.

Point 2: All the species are either outcrossed with wind or insect as pollinators that infers 5 km distance isolation from crops. Gene flow with crops are likely and authors should take into account in the database the isolation of the Natura 2000 sites. I am not Spanish and I cannot judge of this point for so many species.

Response 2:

We understand the reviewer 2 concern on outcrossing events with crops cultivated close to the selected areas. As outcrossing events assessment were not among the objectives of our work, but it is obviously a point to take into account if associations’ populations are intended to be a source of genetic variability for crop breeding, we have included the following sentences (lines 401-402): “Before any genetic reserve is established the risk of outcrossing events with related crop species in surrounding areas should be evaluated and properly addressed”.

Point 3: Adaptation patterns are not so easy to find and I would be convinced that all species in the ELC show adaptation patterns for traits related to climate change. Salinity and drought traits do not share common patterns because drought stresses are very difficult to experiment in wild populations … Either in the introduction or in discussion these points should be added

Response 3: Salinity and drought were only mentioned as an example of the search for trait adaptation to different environments through the use of ELC maps. We are not searching for these adaptation patterns in particular, but only pointing to the selection of associations’ populations with higher probabilities of containing potential genetic diversity of adaptive value. These adaptation patterns should be further assessed by experimental trials relating given ELC categories with specific traits. To avoid confusion to the readers we have deleted the explicit mention to drought and salinity from the text. In addition, we have included in lines 61-63 (revised version, introduction) the following sentence: “In any case, when specific traits are sought, the use of ecogeographic information as a proxy for adaptive genetic differentiation among populations must subsequently be validated by appropriate phenotype evaluation and/or molecular marker characterization of the target populations”.

Point 4: Moreover, these genetic resources remain not evaluated and I cannot find in the text data on on their interest for breeding to climate change, except that there are wild and in Natura 2000 sites. These sites are frail and their evolution with climate change remains uncertain. In their sites the species cannot be evaluated that delays their use for eventual breeding. Climate changes are so rapid that I have doubts about the adaptation of wild species to survive in Natura 2000 sites..… Either in the introduction or in discussion these points should be added

Response 4: We have now included in the discussion the need to properly evaluate these resources through phenotyping in common gardens and/or genetic characterization (lines 279-280). The protection coverage given by N2000 is a good start for the establishment of genetic reserves where target CWR populations can be conserved. Additional measures may be needed to warrant the long-term conservation and population viability through the challenges of climate change. In this sense, we have included in the discussion (lines 405-407) the following sentence: “Finally, these genetic reserves should meet a minimum set of quality standards (Iriondo et al., 2012) to mitigate the effects of climate change and ensure the long-term conservation of these resources”. The reference has been included in the list of references.

Point 5: In fact the authors should develop the benefit of a genetic resources in wild species located in an association versus genetic resources in collection. The interest of the associations is not so clear for me

Response 5: The main interest of using the Asso-EcoU conservation unit is the possibility of managing the conservation of multiple species at the same time and, in addition, considering their potential genetic diversity. This idea is mentioned in various sections and passages: abstract, introduction (lines 100-106), discussion (lines 296-298) and conclusion (lines 411-415). We have now briefly stated the interest of in situ conservation versus ex situ collections in the introduction (lines 46-49).

Point 6: But many cautions should be given and the concept of Asso-EcoU associations maintained in different environments, it should explain the relationships with adaptation patterns.

Response 6: The cautions have now been explicitly stated in the introduction and in the discussion. The relationship between environmental heterogeneity and adaptation patterns has also been better clarified in these sections.

Point 7: Please develop all sigles112 MGRS Military Grid Reference System

Response 7: Done

Point 8: 199 phytosociological associations resulted in 165 unique Asso-EcoUs Are there in fig. 2?

Response 8: Yes, that was a mistake. We have corrected the Fig. 2 caption and also added the number of inventories representing the 165 Asso-EcoUs.

Point 9: 205 change ecogeographical categories

Response 9: Done

Point 10: in my opinion the breeding objectives are too far, I would suggest to Authors to limit their ambition to the concept of Asso-EcoU not in relationships with climate changes because there are too much unknown relationships and that impair the studies

Response 10: We agree with the reviewer 2 view about the uncertainty of responses of plants to climate change. On the other hand, we believe that justification for the use of crop wild relatives as a source of variability lacked by crops and their use in breeding processes is supported by references in the text. The focus of our work is to propose an efficient use of existing plant genetic resources, by the selection of protected sites with populations of CWR species that are likely to contain genetic variability shaped by environmental pressures. Pre-breeding or breeding processes are out of the scope of this paper.

Point 11: the evolution of associations in some sites basing on climate changes, obviously are long term studies... and eventually to introduce in the future studies neutral and linked molecular markers to salt and drought tolerance and other environmental traits to follow the evolution of markers that will indicate the fate of the traits.

Response 11: We agree with this point. We hope that the inclusion of sentences clearly stating that adaptation patterns potentially contained by Asso-EcoUs populations should be experimentally tested (lines 61-63 and lines 401-405) answers this concern.

References:

Iriondo, J.M.; Maxted, N.; Kell, S.P.; Ford-Lloyd, B.V.; Lara-Romero, C.; Labokas, J.; Magos Brehm, J. Quality standards for genetic reserve conservation of crop wild relatives. In Agrobiodiversity Conservation. Securing the diversity of crop wild relatives and landraces; Maxted, N.; Dulloo, E.; Ford-Lloyd, B.V.; Frese, L.; Iriondo, J.;Pinheiro de Carvalho, M.A.A., Eds.; CAB International: Oxfordshire, UK, 2012; pp. 72–77.
